# CHARACTERIZING TRAINABILITY, EXPRESSIVITY, AND GENERALIZATION OF NEURAL ARCHITECTURE WITH METRICS FROM NEURAL TANGENT KERNEL

## ABSTRACT

Zero-shot neural architecture search aims to predict multiple characteristics of neural architectures using proxy indicators without actual training, however, most methods focus on evaluating only a single characteristic of neural networks. Since the Neural Tangent Kernel (NTK) offers a promising theoretical framework for understanding the characteristics of neural networks, we propose NTK-score, a proxy indicator that includes three metrics derived from NTK's eigenvalues and kernel regression, to assess three critical characteristics: trainability, expressivity, and generalization. Moreover, to exploit three metrics of our NTK-score, we employ the Borda Count approach on our NTK-score to rank architectures in neural architecture search. Compared with state-of-the-art proxies, experimental results demonstrate that the NTK-score correlates well with both the test accuracy and training time of architectures, and outperforms comparison proxies across various search spaces and methods, including NAS-bench-201, DARTS, and ResNet, as well as pruning, reinforce, and evolutionary algorithm.

## 1 INTRODUCTION

Neural networks have brought many important technological breakthroughs and innovations to the field of computer vision, promoting the rapid development and widespread application of this field. However, manually designing neural network architectures is challenging and requires extensive professional knowledge and experience, as well as a lot of experiments and adjustments Baker et al. (2016), Rumiantsev & Coates (2023). Therefore, automatically designing neural networks, such as Neural Architecture Search (NAS), has attracted increasing research interest.

NAS enables automated neural network design by searching the space of possible network architectures and evaluating the performance of each architecture. As more and more NAS methods have been proposed, the NAS methods can now be categorized into three types based on training frequency: multi-shot NAS Xie & Yuille (2017), one-shot NAS Liu et al. (2018) and zero-shot NAS Chen et al. (2021). This categorizing reflects the varying consumption of time and resources involved in training. Due to the huge overhead of time and resources required for training architectures, our work focuses on the zero-shot NAS.

Zero-shot NAS evaluates and selects potential neural network architectures based on some sophisticated metrics, bypassing network training. The lack of training necessitates an accuracy ranking agent Chen et al. (2021), Chen et al. (2023b). For instance, TEG-NAS Chen et al. (2023b) and AZ-NAS Lee & Ham (2024) employ their proxies to rank the neural networks based on multiple characteristics. However, most proposed agents focus solely on one characteristic, often leading to results that fail to outperform certain naive agents Li et al. (2024). As Neural Tangent Kernel (NTK) provides a stable mathematical framework to understand and analyze the characteristics of neural networks by linearizing the training dynamics and remaining constant over time in the infinite-width limit, our designed agent utilizes NTK to evaluate neural networks across the dimensions of trainability, expressivity, and generalization simultaneously.

The trainability of a neural network refers to how quickly it converges to the expected loss or accuracy during training, indicating its ability to adapt to the training data Li et al. (2024). To characterize the trainability, TE-NAS Chen et al. (2021) utilizes the condition number $\kappa$ Xiao et al. (2020), which

only focuses on the *max* and *min* eigenvalues, somewhat ignores the distribution of eigenvalues, as the entire spectrum of the NTK can be a better measure Wang et al. (2023). We argue that neural networks with better generalization capabilities tend to exhibit more similar NTK eigenvalues. Therefore, we present a metric that quantifies an architecture's trainability by utilizing the ratio of larger NTK eigenvalues.

Expressivity of a neural network refers to its ability to capture and represent the number of complex patterns and relationships within data Raghu et al. (2017). A common expressivity measure indicator is the number of parameters, which is not applicable to deep networks as networks with the same number of parameters but different depths perform quite differently. Hence, the current NAS approaches mostly use the Number of Linear Regions (NLR) divided by the ReLU function Mellor et al. (2021), Chen et al. (2021) rather than NTK. Considering that the output of a neural network can be efficiently evaluated using NTK Lee et al. (2019), and the network's capacity to map similar inputs to distinct regions can serve as a reflection of the network's expressive ability Xiong et al. (2020), we propose an expressivity metric by computing the cross-entropy of the output distributions of two similar inputs, in which the output distributions are calculated through NTK.

Generalization of a neural network also stands as a crucial indicator for evaluating the neural network's capacity to operate effectively on unseen data Zhu et al. (2022). Recent research has associated NTK with the generalization capabilities of neural networks, such as degree $k$ fractional variance of the NTK kernel Yang & Salman (2019) and Mean Squared Error (MSE) loss in NTK kernel regression Chen et al. (2023a). NTK kernel regression simply represents the network's label on the test set Lee et al. (2019), and its loss with the true label shows the network's ability on unseen data. We also use this metric to evaluate the generalization ability of neural networks.

After characterizing the trainability, expressivity, and generalization of a neural network based on NTK, we organize the three values of these characteristics as a proxy NTK-score for ranking neural architectures in NAS. Due to the disparate nature of these three score values, direct arithmetic operations are challenging. To address this, we employ the Borda Count approach on the NTK-score's three characteristics, and design an NAS framework called SABoC-NAS.

In summary, our contributions are as follows:

- To evaluate and rank neural architectures, we introduce NTK-score, which includes three training-free metrics derived from NTK's eigenvalues and kernel regression, to assess the trainability, expressivity, and generalization of a neural network respectively.

- We develop a NAS framework using our NTK-score, namely SABoC-NAS, managing trade-offs among trainability, expressivity, and generalization by Borda Count.

## 2 RELATED WORK

### 2.1 NEURAL ARCHITECTURE SEARCH

At first, people used brute force methods to directly use the accuracy of the trained network architecture for screening, including Genetic CNN Xie & Yuille (2017) and MetaQNN Baker et al. (2016). Then, in order to reduce the overhead required for training, one-shot was proposed, that is, Supernet is trained only once, and multiple different networks are obtained through weight sharing, such as DARTS Liu et al. (2018), FBNet Wu et al. (2019), GreedyNAS You et al. (2020) and Single-Path One-Shot NAS Guo et al. (2020).

Recently, zero-shot NAS becomes the mainstream method of NAS Li et al. (2024). Gradient of deep network parameters is first proposed to design agents that can rank the accuracy of candidate network architectures, such as Fisher Liu et al. (2021), SNIP Lee et al. (2018), Synflow Tanaka et al. (2020), GraSP Wang et al. (2020), Gradnorm Abdelfattah et al. (2021), ZiCo Bhardwaj et al. (2023).

TE-NAS Chen et al. (2021) ranked architectures by analyzing the spectrum of NTK and the number of linear regions in the input space that respectively imply the trainability and expressivity of the neural network. On the basis of TE-NAS, TEG-NAS Chen et al. (2023a) completed the training indicators in generalization and promotes the visualization of the search space. KNAS Xu et al. (2021) found a practical gradient kernel that exhibits strong correlations with both training loss and validation performance and proposed a new kernel based architecture search approach. Further-

more, Label-Gradient Alignment (LGA) Mok et al. (2022) is introduced as a metric based on NTK, to capture the extensive nonlinear characteristics present in contemporary neural architectures. Additionally, Neural Network Gaussian Process (NNGP) Rumiantsev & Coates (2023) which can be computed more efficiently was introduced as a kernel metric to evaluate the architectures faster.

What's more, recent NAS research is increasingly focusing on evaluating one or more specific capabilities of the network. Zen-NAS Lin et al. (2021) directly searched high expressivity architectures in a data-free manner by maximizing the target network's Zen-Score for a given inference budget. In addtion, gradient Signal-To-Noise Ratio (GSNR) Sun et al. (2023) which has been shown to correlate with the generalization performance of neural networks, have been used as a zero-Shot NAS agent to predict network accuracy upon initialization. SWAP-NAS Peng et al. (2024) presented Sample-Wise Activation Patterns and its derivative SWAP-score to measure the architectures' expressivity, which can be further enhanced by regularization. AZ-NAS Lee & Ham (2024) proposed a zero-shot proxy that evaluates architectures along four complementary dimensions: expressiveness, progressiveness, trainability and complexity, which can be evaluated simultaneously in a single forward and backward pass, as well as their nonlinear ranking aggregation method.

## 2.2 NEURAL TANGENT KERNEL

The concept of NTK was first proposed to prove that the gradient descent of artificial neural networks is equivalent to kernel gradient descent. Further research shows that NTK enables the scrutiny of the network's trainability, expressivity, and generalization.

**Trainability.** The condition number $\kappa = \lambda_{max}/\lambda_{min}$ and the largest/smallest eigenvalue of the NTK $\lambda_{max/min}$ Xiao et al. (2020) are used to analyze trainability. The degree k fractional variance Yang & Salman (2019) is proposed as a metric to evaluate the generalization properties of neural networks. In addition, the training process of a neural network can be decomposed along different directions defined by the eigenfunctions of the neural tangent kernel, each direction having its own convergence rate determined by the corresponding eigenvalues Cao et al. (2019), Bowman & Montúfar (2022). Furthermore, it is borne out that larger eigenvalues express the convergence speed and the learning rate is related to the eigenvalues Kopitkov & Indelman (2020).

**Expressivity.** The dynamics of the network function $f_\theta$ aligns with kernel gradient descent in function space concerning a limiting kernel during training Jacot et al. (2018). Moreover, the ODE of the neural network output with respect to NTK is obtained Lee et al. (2019). The result is further extended by decomposing the ODE along different eigenvectors Xiao et al. (2020). Since the output can be easily represented through NTK, NTK can be used to analyze the expressivity of neural networks.

**Generalization.** NTK can provide memory, optimization and generalization guarantees in deep neural networks Bombari et al. (2022). The minimum eigenvalue is used to establish the generalization error bound in stochastic gradient descent training Zhu et al. (2022), Zhu et al. (2023). The positive definiteness of NTK is proved by providing the lower bound of the minimum eigenvalue of NTK in deep learning theory, both in the limiting case of infinite widths and for finite widths Nguyen et al. (2021), Bombari et al. (2022), Zhu et al. (2022), Zhu et al. (2023), Banerjee et al. (2023).

## 3 PRELIMINARY

Given a neural network $f$, NTK at time $t$ is defined as an $n \times n$ positive semidefinite matrix $H_t$ whose $(i,j)th$-entry is $< \frac{\partial f(\theta_t, x_i)}{\partial \theta}, \frac{\partial f(\theta_t, x_j)}{\partial \theta} >$, where $f(\theta_t, x)$ is the output of the network, $\theta_t$ is all parameters of the network and $x$ is the input.

The evolving output $f(x)$ of the neural network over time Lee et al. (2019) can be represented by Eq. (1) and Eq. (2)

$$f(X_{train}) = (I - e^{-\eta H_{train,train}t})Y_{train}, \tag{1}$$

$$f(X_{test}) = H_{test,train}H_{train,train}^{-1}(I - e^{-\eta H_{train,train}t})Y_{train}, \tag{2}$$

where $H_{train,train}$ is NTK calculated on the training dataset at initialization, $H_{test,train}$ is NTK calculated on the training and test dataset at initialization, $X_{train}$ is the training data, $X_{test}$ is the test data, $Y_{train}$ is the labels of training data.

Neglecting the time factor $t$, the equation can be simplified calculated by Eq. (3)

$$f(X_{test}) = H_{test,train}H_{train,train}^{-1}Y_{train}, \tag{3}$$

enabling us to efficiently compute the simple output of the neural network without training.

The equation is further decomposed $H_0$ along different eigenfunctions Xiao et al. (2020), evolve as Eq. (4)

$$f(X_{train})_i = (I - e^{-\eta\lambda_i t})Y_{train,i}, \tag{4}$$

where $\lambda_i$ is the eigenvalues of $H_0$ and maximum feasible learning rate $\eta \sim 2/\lambda_0$ Lee et al. (2019).

During the training phase of the neural network, decomposition along distinct directions defined by the eigenfunctions of the neural tangent kernel reveals unique convergence rates which are dictated by the corresponding eigenvalues Cao et al. (2019), Bowman & Montúfar (2022). Therefore, the condition number of NTK, defined as $\kappa = \frac{\lambda_{max}}{\lambda_{min}}$ is introduced and used as a metric to quantify the trainability of the neural network by TE-NAS Chen et al. (2021), which regards that a neural network is not trainable if $\kappa$ diverges.

## 4 METHOD

After defining NTK-score as a triplet to characterize trainability, expressivity, and generalization of a neural architecture, we depict a training-free framework SABoC-NAS integrated Borda Count approach for ranking neural architectures.

### 4.1 NTK-SCORE ON TRAINABILITY, EXPRESSIVITY, AND GENERALIZATION

NTK-score, denoted by a triplet $(S_t, S_e, S_g)$, describes a neural architecture across three dimensions: trainability, expressivity, and generalization, respectively. All elements of NTK-score are derived from Neural Tangent Kernel (NTK) of the given architecture, and explained as follows.

**Trainability**. Given a neural architecture $f$, a training dataset $X_{train}$, the NTK $H_{train,train}$ Is computed on the training dataset, and the value of trainability metric $S_t$ is calculated by Eq. (5)

$$S_t = \frac{\sum_{i=0}^{\lceil\sqrt{n}\rceil-1}\lambda_i}{\sum_{i=0}^{n-1}\lambda_i}, \tag{5}$$

where $\lambda_0 > \lambda_1 > ... > \lambda_{n-1}$ are the eigenvalues of the NTK $H_{train,train}$, $n$ is the batch size of the input data, and $\lceil\sqrt{n}\rceil$ denotes a small fraction of larger eigenvalues which accounts for most of the sum of all eigenvalue.

**Expressivity**. To characterize the expressivity of a neural architecture, we use both of test dataset $X_{test}$ and training dataset $X_{train}$, and let the NTK $H_{test,train}$ be on training dataset and test dataset, the output of the neural network on the test dataset $f(X_{test})$ is calculated by Eq. (6)

$$f(X_{test}) = H_{test,train}H_{train,train}^{-1}Y_{train}, \tag{6}$$

where $Y_{train}$ is the label set of training dataset. Eq. (6) represents a NTK kernel regression.

And then, we apply a minor perturbation $\epsilon$ to $X_{test}$, yielding $X_{test'}$, and subsequently compute the output $f(X_{test'})$ in the same way by Eq. (7) and Eq. (8)

$$X_{test'} = X_{test} + \epsilon, \ \epsilon \sim \mathcal{N}(0, 10^{-4}) \tag{7}$$

$$f(X_{test'}) = H_{test',train}H_{train,train}^{-1}Y_{train}. \tag{8}$$

having $f(X_{test})$ and $f(X_{test'})$, we calculate the difference between them as the value of expressivity metric $S_e$ by Eq. (9)

$$S_e = -CrossEntropy(f(X_{test}), f(X_{test'})), \tag{9}$$

where $-$ is unify the standard so that the smaller the $S_e$, the stronger the expressivity.

**Generalization**. Following TEG-NAS Chen et al. (2023a), we use the square loss between the succinctly estimated output $f(X_{test})$ and the true label $Y_{test}$ of test dataset as the value of generalization metric $S_g$, as shown by Eq. (10)

$$S_g = ||f(X_{test}) - Y_{test}||_2. \tag{10}$$

## 4.2 SABoC-NAS: Searching Architecture by Borda Count

We depict SABoC-NAS framework for selecting the target architecture with high scores across all three metrics, as it is proved that there is no single architecture that is optimal in all three characteristics given a fixed budget Chen et al. (2023b). Considering the significant disparity in the magnitudes of the three metrics, simple addition or subtraction is inadequate for fusing them, we employ the Borda Count approach to flexibly trade-off between trainability, expressivity, and generalization.

Specifically, given a set of architectures $\{a_1, a_2, \ldots, a_m\}$, we calculate the NTK-scores of all $m$ architectures separately, denoted by Eq. (11)

$$S_i^{1:m} = \{S_i^1, S_i^2, \ldots, S_i^n\}, \qquad i = t, e, g, \tag{11}$$

where subscript $i$ of metric value $S_i^k$ indicates different metrics, and superscript $k$ indicates architecture $a_k$.

Having the three metric values of all architectures $S_t^{1:m}, S_e^{1:m}, S_g^{1:m}$, we sort all architectures based on each metric's values separately, and calculate the Borda Count rankings, as shown by Eq. (12)

$$S_r^{1:m} = rank(S_t^{1:m}) + rank(S_e^{1:m}) + rank(S_g^{1:m}), \tag{12}$$

where $rank(\cdot)$ sorts input values and returns a permutation of $\{1, 2, \ldots, m\}$ which indicates the positions of all input values at the ordered array.

The core component of our SABoC-NAS framework is the proxy $S_r^{1:m}$ for ranking architectures, and various search algorithms, such as pruning algorithm, reinforce algorithm and evolutionary algorithm, could be easily integrated into our SABoC-NAS framework.

## 5 Experiment

We demonstrate the correlation coefficients between our training-free NTK-score and established measurement indicators, along with other SOTA proxies (Sec.5.2), and conduct an ablation study to explore different combinations of the NTK-score and analyze each component (Sec.5.3). In addition, we compare our SABoC-NAS with the SOTA zero-shot proxies in terms of accuracy and search cost using different search spaces(Sec.5.4) and search methods (Sec.5.5).

### 5.1 Implementation Details

We introduce the search space, search methods, SOTA proxies, as well as parameter settings used in the experiments.

**Search Space.** NAS-Bench-201 Dong & Yang (2020) search space contains 5 operations: none (zero), skip connection, 1 x 1 convolution, 3 x 3 convolution, and average pooling 3 x 3.

DARTS Liu et al. (2018) search space contains 8 operations: none (zero), skip connection, separable convolution $3 \times 3$ and $5 \times 5$, dilated separable convolution $3 \times 3$ and $5 \times 5$, max pooling $3 \times 3$, average pooling $3 \times 3$.

ResNet He et al. (2016) search space consists of residual blocks and bottleneck blocks. The convolution kernel size is in the set $\{3,5,7\}$

**Search Method.** Pruning Algorithm, slimier to TE-NAS Chen et al. (2021). The neural network is structured with standardized cells of parallel edges. For an operation performed on a parallel edge between two cells in each iteration $t$, the NTK-scores of the network are calculated both before and after the operation. The pruning probability is then determined by comprehensively evaluating the scores in three characteristics: trainability, expressivity, and generalization. The algorithm iteratively prunes one of the parallel edges until only a single path remains.

Reinforce Algorithm, slimier to TEG-NAS Chen et al. (2023a). The action space in reinforcement learning is defined as the edge operation between cells, and the reward is defined as the comprehensive score of the trainability, expressivity, and generalization of the new architecture generated by the selected action. The algorithm selects actions from the action space according to probability, and then updates the probability of action selection according to the reward. The above operation is repeated for $T$ steps, and finally the top ranked architecture by Borda Count is selected.

Evolutionary Algorithm, slimier to Zen-NAS Lin et al. (2021). In each iteration $t$, a new architecture is generated through genetic operations and mutations, which is then added to the population. When the population size exceeds the maximum limit, the architecture with the worst score for each of the three metrics is removed, resulting in the simultaneous removal of three architectures. At the conclusion of iterations, we select the target architecture based on the comprehensive ranking by the Borda Count of the three metrics.

**Dataset.** CIFAR-10 and CIFAR-100 Krizhevsky et al. (2009) are both widely used benchmark datasets in the field of computer vision. CIFAR-10 consists of 60,000 32x32 color images in 10 classes, with 6,000 images per class. CIFAR-100 is similar but contains 100 classes, with each class containing 600 images.

ImageNet-16-120 Chrabaszcz et al. (2017) is a subset of the ImageNet dataset, specifically curated for benchmarking purposes. It consists of 16 object categories with a total of 120 fine-grained classes. Each category contains a varying number of classes, with a total of 1,281 images for training and 50 images for validation per class.

**SOTA Proxies.** Gradient-based methods such as Fisher Liu et al. (2021), SNIP Lee et al. (2018), Synflow Tanaka et al. (2020), GraSP Wang et al. (2020), Gradnorm Abdelfattah et al. (2021), ZiCo Li et al. (2023)

Kernel-based methods such as ETE-NAS Rumiantsev & Coates (2023), KNAS Xu et al. (2021), LGA Mok et al. (2022), TE-NAS Chen et al. (2021), TEG-NAS Chen et al. (2023a)

Other methods such as NASWOT Mellor et al. (2021), Zen-NAS Lin et al. (2021), SWAP-NAS Peng et al. (2024), AZ-NAS Lee & Ham (2024).

**Parameter Settings.** We set batch size = 64 and use Kaiming normal initialization $\mathcal{N}(0, N_l)$ to initialize the network, where $N_l$ is the width at layer $l$.

## 5.2 NTK-SCORE VS SOTA PROXIES

To demonstrate the effectiveness of the NTK-score, we calculate the Kend-$\tau$ correlation coefficient between the predicted rankings derived from various zero-shot metrics and the actual rankings based on established measurement indicators Dong & Yang (2020) for CIFAR-100 in NAS-Bench-201.

We calculates the Kend-$\tau$ correlation coefficient between the predicted rankings derived from the NTK-score and other state-of-the-art (SOTA) proxies, compared with the actual rankings based on Test Set Accuracy for CIFAR-100 in NAS-Bench-201, as we consider classification accuracy to be the most critical indicator of architecture performance. Moreover, we use the 100-epoch Training Accuracy as an auxiliary measure of trainability, as it reflects the model's training speed. A higher 100-epoch training accuracy indicates that fewer epochs, and thus less time, are required to reach the specified target accuracy. The results are presented in Figure 1, where the *x*-axis represents the correlation coefficient with Test Set Accuracy, and the *y*-axis represents 100-epoch Training Accuracy, derived from 1024 randomly selected architectures.

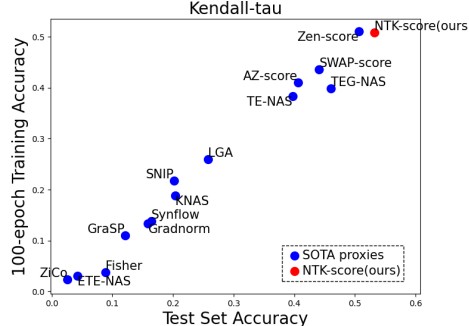

Figure 1: Kend-$\tau$ Correlation Coefficient of various metrics on CIFAR-100 of NAS-Bench-201

The correlation coefficient between NTK-score and Test Set Accuracy reaches 0.532, representing the highest correlation observed, while the correlation coefficient with 100-epoch Training Accuracy is 0.509, which is also the second highest, second only to Zen-score Lin et al. (2021) of 0.551. This demonstrates that the NTK-score is strongly linked to both model accuracy and training time,

positioning it as a valuable metric for ranking architectures. By leveraging the NTK-score, we can more effectively identify models that achieve higher accuracy while minimizing training time.

Table 1: Ablation Study of Kend-$\tau$ Correlation Coefficient with Test Set Accuracy on CIFAR-100 of NAS-Bench-201

| Methods | Kend-$\tau$ |
|---|---|
| $S_t$ | 0.433 |
| $S_e$ | 0.514 |
| $S_g$ | 0.513 |
| $S_t$ & $S_e$ | 0.524 |
| $S_t$ & $S_g$ | 0.522 |
| $S_e$ & $S_g$ | 0.518 |
| $S_t$ & $S_e$ & $S_g$ | 0.532 |

Table 2: Kend-$\tau$ Correlation Coefficient with Test Set Accuracy of various batch size of $S_t$ on CIFAR-100 of NAS-Bench-201

| batch size $n$ | Kend-$\tau$ |
|---|---|
| 16 | 0.324 |
| 32 | 0.401 |
| 64 | 0.433 |
| 96 | 0.459 |
| 128 | 0.477 |

### 5.3 ABLATION STUDY

To further validate the effectiveness of each metric of NTK-score, we examine the Kend-$\tau$ correlation coefficient using various combinations of $S_t$, $S_e$, and $S_g$ in the NTK-score. Table 1 displays the correlation coefficients between the predicted rankings derived from different NTK-score combinations and the actual rankings for Test Set Accuracy on CIFAR-100 from NAS-Bench-201.

When using only one metric, $S_e$ achieves the highest correlation coefficient of 0.514. When employing two metrics, $S_t$&$S_e$ has the highest correlation coefficient at 0.522, while the lowest, $S_e$&$S_g$, was still significant at 0.518. It is evident that using a composite ranking based on multiple metrics results in higher relevance compared with using a single metric. When all three characteristics of the NTK-score are combined, the relevance reaches its maximum at 0.532.

In addition, we also analyze the each component of NTK-score. Figure 2 shows the correlation coefficient between each component of NTK-score and Test Set Accuracy.

**Trainability.** As shown in Figure 2a, the Kend-$\tau$ reaches 0.433, indicating that $S_t$ correlates with Test Set Accuracy and can be used to effectively filter out architectures. In comparison, the condition number $\kappa$ used in TE-NAS Chen et al. (2021) yields a Kend-$\tau$ of 0.397, showing that $S_t$ provides an improvement of 0.036, validating the effectiveness of our approach.

Analyzing the reasons for the improvement, the neural network can be decomposed along various eigenfunctions, each of which is associated with a different eigenvalue. And the output equation of the neural network decomposing along eigenfunction with larger corresponding eigenvalue will tend to stabilize faster. Furthermore, a network's convergence primarily relies on a subset of its eigenfunctions, which is why we use $\sqrt{n}$ in the ratio. Therefore, networks with a tighter distribution of eigenvalue values converge faster, making $S_t$, which utilizes the eigenvalue ratio, more effective than $\kappa$, which may overlook the distribution of eigenvalues when assessing this feature.

As $S_t$ relies on the batch size $n$, we further calaulate Kend-$\tau$ correlation coefficient of different $n$ for $S_t$ to verify the effectiveness of $S_t$ depicted in Table 2. As $n$ increases, Kend-$\tau$ of $S_t$ continues to rise. This trend is reasonable because a larger batch size leads to a more complex computation of NTK, allowing $S_t$ to become a more precise measure. Consequently, Kend-$\tau$ increases, reaching its highest value of 0.477 at a batch size of 128. Even with $n = 16$, Kend-$\tau$ remains significant at 0.324.

**Expressivity and Generalization.** As shown in Figures 2b and 2c, both $S_e$ and $S_g$ exhibit a strong correlation with Test Set Accuracy, with Kend-$\tau$ values of 0.514 and 0.513, respectively.

The expressive capacity of a neural network can be evaluated by the number of divided linear regions, and since NTK transforms a neural network into kernel regression, the difference in outputs for similar inputs serves as a useful metric for assessing the network's expressive capability. A higher $S_e$ indicates the network's enhanced ability to differentiate between similar samples, showcasing its proficiency in capturing and representing complex patterns and relationships within the data, thus leading to higher accuracy.

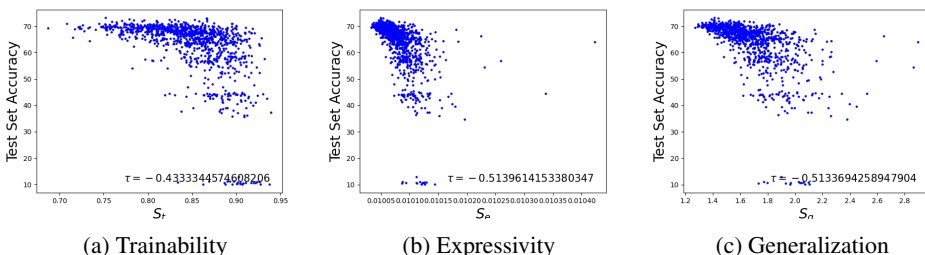

|  (a) Trainability  |  (b) Expressivity  |  (c) Generalization  |

Figure 2: NTK-score Evaluating on CIFAR-100 from NAS-Bench-201.

For generalization, it is intuitive that to be evaluated using the loss of the initialized network output and the true label, as the training process of the neural network is to minimize the loss between the predicted output and the true label by continuously adjusting the weights.

## 5.4 NTK-SCORE USED IN DIFFERENT SEARCH SPACES

In this section, we employ widely-used kernel methods alongside an efficient pruning algorithm for our search approach, utilizing NAS-Bench-201 and DARTS as the primary search spaces, running five trials with different random seeds. For architectures derived from DARTS, we conduct training over 400 epochs to assess accuracy.

To further evaluate the NTK-score's performance in more complex search spaces, we also explore the ResNet architecture, known for its capability to construct intricate networks with high performance. Following the approach of Zen-NAS Lin et al. (2021), we implement an evolutionary algorithm with a population size of 256 and conduct 24,000 evolutionary iterations.

Due to the differing search spaces and methods employed in the original papers for various SOTA metrics, we adopt a unified setup for our experiments to ensure fairness. All results are reproduced using the official code provided by the authors.

Table 3: Pruning Results on NAS-Bench-201

| Methods | CIFAR-10 | CIFAR-100 | ImageNet-16-120 | Search Cost(s) |
|---|---|---|---|---|
|  | Accuracy-1 | Accuracy-1 | Accuracy-1 |  |
| Fisher | 90.86 | 66.67 | 37.50 | 109 |
| SNIP | 91.91 | 67.34 | 39.18 | 92 |
| Synflow | 93.43 | 70.42 | 42.88 | 82 |
| GraSP | 93.11 | 70.21 | 43.66 | 93 |
| Gradnorm | 92.01 | 67.27 | 39.59 | 49 |
| ZiCo | 93.28 | 70.58 | 43.60 | 50 |
| Zen-NAS | 93.46 | 70.36 | 43.25 | 20 |
| SWAP-NAS | 93.18 | 70.14 | 42.09 | 18 |
| AZ-NAS | 93.01 | 70.40 | 44.68 | 26 |
| ETE-NAS | 92.75 | 69.94 | 41.38 | 38 |
| KNAS | 93.03 | 70.22 | 42.61 | 452 |
| LGA | 93.16 | 69.95 | 44.13 | 434 |
| TE-NAS | 93.31 | 70.38 | 44.53 | 1964 |
| TEG-NAS | 93.20 | 70.48 | 44.68 | 3228 |
| SABoC-NAS(ours) | **93.63** | **71.06** | **45.10** | 921 |

### 5.4.1 PRUNING ON NAS-BENCH-201

Table 3 displays the top-1 accuracy and search cost of architectures generated using the pruning algorithm on NAS-Bench-201 for CIFAR-10, CIFAR-100, and ImageNet-16-120. SABoC-NAS achieves the highest accuracy across all three datasets, with improvements of at least 0.17%, 0.48%, and 0.42%, respectively.

Table 4: Pruning Results on DARTS

| Methods | CIFAR-10 | CIFAR-100 | Search Cost(s) |
|---|---|---|---|
| | Accuracy-1 | Accuracy-1 | |
| ETE-NAS | 95.36 | 80.08 | 1793 |
| KNAS | 95.84 | 79.47 | 13992 |
| LGA | 96.25 | 80.58 | 14499 |
| TE-NAS | 96.50 | 80.34 | 25169 |
| TEG-NAS | 96.62 | 81.76 | 25004 |
| SABoC-NAS(ours) | **96.81** | **81.82** | 25232 |

Table 5: Evolution Results on ResNet

| Methods | CIFAR-10 | | CIFAR-100 | | Search Cost(h) |
|---|---|---|---|---|---|
| | Accuracy-1 | Accuracy-5 | Accuracy-1 | Accuracy-5 | |
| Zen-NAS | **97.28** | **99.93** | **81.67** | **96.17** | 11.6 |
| NASWOT | 95.14 | 99.83 | 71.64 | 91.46 | 16.7 |
| ETE-NAS | 95.04 | 99.85 | 73.73 | 93.65 | 62.3 |
| TE-NAS | 96.03 | 99.90 | 76.44 | 94.10 | 65.0 |
| TEG-NAS | 96.46 | 99.94 | 78.51 | 95.02 | 124.7 |
| SABoC-NAS(ours) | 96.97 | 99.92 | 80.11 | 95.97 | 137.0 |

In comparison to methods that rely on forward propagation or gradients, which generally incur lower computational costs, SABoC-NAS demonstrates an average enhancement of 0.94%, 1.79%, and 3.27% on CIFAR-10, CIFAR-100, and ImageNet-16-120, respectively. Additionally, when evaluated against NTK-based methods, SABoC-NAS shows an average improvement of 0.46%, 0.8%, and 1.11%, indicating that the NTK-score serves as a more effective metric.

Notably, SABoC-NAS leverages NTK to simplify the complex expressivity calculations associated with the number of linear regions, significantly reducing search costs. This stands in contrast to TEG-NAS Chen et al. (2023a), which also evaluates architectures based on NTK.

### 5.4.2 PRUNING ON DARTS

Table 4 shows the top-1 accuracy and search cost of architectures generated using the pruning algorithm on DARTS for CIFAR-10 and CIFAR-100. We compare several relevant NTK kernel methods, including KNAS Xu et al. (2021) using Frobenius norm, LGA Mok et al. (2022) using Label-Gradient Alignment, TE-NAS Chen et al. (2021) leveraging $\kappa$, TEG-NAS Chen et al. (2023a) employing both $\kappa$ and $MSE$, and ETE-NAS Rumiantsev & Coates (2023) utilizing NNGP.

Although SABoC-NAS requires more time than NNGP, it generates architectures with superior accuracy, notably achieving an improvement of 1.74% on CIFAR-100. Furthermore, when compared with NTK-based methods, SABoC-NAS yields the highest accuracy, with average enhancements of 0.51% on CIFAR-10 and 1.28% on CIFAR-100, underscoring the effectiveness of the NTK-score.

### 5.4.3 EVOLVING ON RESNET

Table 5 presents the top-1 and top-5 accuracy, along with the search cost of architectures generated for CIFAR-10 and CIFAR-100. SABoC-NAS outperforms other kernel-based methods while maintaining a comparable search time, achieving the most significant improvement in top-1 accuracy on the CIFAR-100 dataset, with an average increase of 3.88%.

Additionally, SABoC-NAS ranks second only to Zen-NAS Lin et al. (2021), which is specifically designed for ResNet architectures and is less effective for DARTS and NAS-Bench-201. This suggests that our NTK-score is well-suited for more complex and contemporary architectures, delivering excellent performance.

When compared with TE-NAS Chen et al. (2021) and TEG-NAS Chen et al. (2023a), both of which evaluate architectures based on multiple characteristics, SABoC-NAS demonstrates superior perfor-

mance. We attribute this to the limitations inherent in their methods for calculating the number of linear regions, which hinder their effectiveness in assessing the expressiveness of architectures. In contrast, our NTK-score effectively addresses this issue.

## 5.5 NTK-SCORE USED IN DIFFERENT SEARCH METHODS

To demonstrate the versatility of the NTK-score across various search methods, we test it on NAS-Bench-201 using the reinforce algorithm and evolutionary algorithm, in addition to the pruning algorithm. For comparison, we include AZ-NAS Lee & Ham (2024) and TEG-NAS Chen et al. (2023a), both of which evaluate architectures based on the same three characteristics with identical setups. Each method is run five times with different random seeds.

As shown in table 6, SABoC-NAS achieves the highest accuracy in most scenarios. Notably, compared with AZ-NAS Lee & Ham (2024), SABoC-NAS demonstrates superior performance, particularly on ImageNet-16-120, with a maximum improvement of 4.14%, despite requiring more time. In comparison to TEG-NAS Chen et al. (2023a), which also utilizes kernel methods, SABoC-NAS improves accuracy by an average of 0.36% while reducing computation time by 14.8%. Overall, these results affirm that the NTK-score is applicable to a variety of search methods.

Table 6: Results in different search methods on NAS-Bench-201

| Methods | | CIFAR-10 Accuracy-1 | CIFAR-100 Accuracy-1 | ImageNet-16-120 Accuracy-1 | Search Cost(s) |
|---|---|---|---|---|---|
| reinforce | AZ-NAS | **93.64** | 70.43 | 41.65 | 77 |
| | TEG-NAS | 93.21 | 70.42 | 44.88 | 3885 |
| | SABoC-NAS(ours) | 93.56 | **70.68** | **45.31** | 3058 |
| evolution | AZ-NAS | 93.05 | 69.37 | 40.69 | 279 |
| | TEG-NAS | 93.00 | 70.10 | 44.45 | 9376 |
| | SABoC-NAS(ours) | **93.43** | **70.42** | **44.83** | 8600 |

## 6 CONCLUSION

In this work, we introduce the NTK-score, a metric that leverages NTK to evaluate neural networks across three key characteristics: trainability, expressivity, and generalization. We also present the SABoC-NAS framework, which utilizes the Borda Count approach to effectively integrate the diverse aspects of the NTK-score. By focusing exclusively on eigenvalues and kernel regression derived from the NTK, our method achieves higher accuracy and lower computational costs compared with other kernel-based approaches. In the future, we will place greater emphasis on the theoretical analysis of NTK-based metrics, explore additional applications of NTK in NAS, and conduct more extensive experimental validations.

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

# A   APPENDIX

