# OpenReview forum: "Characterizing trainability, expressivity, and generalization of neural architecture with metrics from neural tangent kernel"
_ICLR.cc/2025/Conference — ICLR 2025 Conference Withdrawn Submission_

### Official Review · Reviewer_vTp9 · 2024-10-28

**Soundness:** 3
**Presentation:** 2
**Contribution:** 2
**Rating:** 3
**Confidence:** 3

**Summary:**

This paper tackles zero-shot neural architecture search (NAS) by proposing NTK-score, a proxy indicator derived from Neural Tangent Kernel that aims to evaluate trainability, expressivity, and generalization simultaneously. While the use of NTK for architecture evaluation is not entirely novel, the authors present an interesting combination of metrics based on eigenvalues and kernel regression, implemented in their SABoC-NAS framework using the Borda Count approach. Overall, the work represents an incremental yet potentially useful contribution to zero-shot NAS.

**Strengths:**

Below is the list of the strong points identified in this work:
- The multi-metric approach to zero-shot NAS using NTK-derived metrics for trainability, expressivity, and generalization offers a more holistic lens on architecture quality than many current approaches, which often focus on a single characteristic.
- The authors present a comparison of NTK-score’s performance relative to state-of-the-art zero-shot NAS proxies, enhancing the credibility of the claimed performance improvements.
- Despite the incremental nature of this work, the NTK-score, if reliable, could add value to NAS applications by potentially reducing the need for exhaustive training during architecture selection.

**Weaknesses:**

Below is the list of weaknesses that I would like to see refuted or clarified by the authors:
- The paper lacks rigorous theoretical justification for using NTK eigenvalues to represent trainability, expressivity, and generalization. While NTK has shown potential in understanding neural dynamics, the extension of its eigenvalues to represent these three qualities lacks a robust mathematical grounding and risks oversimplification. This limits the conceptual soundness of NTK-score.
- The computational demands of NTK-score’s calculations appear unreasonably high for the claimed scalability in large search spaces.
- The reliance on the Borda Count method for metric aggregation is questionable and lacks sufficient justification. Borda Count may be inadequate for effectively capturing the balance across diverse metrics, especially without empirical evidence showing that it improves selection accuracy over simpler or more nuanced methods, such as weighted combinations or Pareto optimisation.
- The definitions of trainability, expressivity, and generalization are operationalised mathematically without offering sufficient interpretative clarity.
- The approach, while technically promising, is not sufficiently validated to be considered generalisable.The Kend-τ Correlation Coefficient is only shown for CIFAR-100 on NASBench-201, when all meta-data of CIFAR-10 and ImageNet16-120 are also present in the search space.
- The writing is frequently repetitive, poorly phrased, and exhibits several structural weaknesses that obscure the main arguments and complicate readability. The introductory section, in particular, appears as a series of loosely connected assertions rather than a cohesive argument or structured exposition.
- There is no graphical abstract.
- The transition from zero-shot NAS to NTK lacks sufficient depth. The introduction asserts that networks with better generalisation capacities tend to have more similar NTK eigenvalues but offers no evidence or citation for this claim.

**Questions:**

Authors are requested to clarify or make changes, as appropriate, based on what is discussed in the ‘Weaknesses’ section.

---

### Official Review · Reviewer_URjv · 2024-11-03

**Soundness:** 2
**Presentation:** 3
**Contribution:** 1
**Rating:** 3
**Confidence:** 5

**Summary:**

This paper proposes a modification of the Neural Tangent Kernel (NTK) based zero-cost proxies for neural architecture search (NAS). Looking at a neural network architecture from the lens of trainability, expressivity and generalization, which are computed using the NTK matrix, a proxy that combines these quantitative values is used to rank various CNN architectures. The authors conduct empirical evaluations on 3 standard search spaces: the DARTS search space, NAS-Bench-201 and a ResNet space, demonstrating the capabilities of their proposed proxy to rank architectures and be used inside a search routine.

**Strengths:**

I think the paper is well-written, covering the necessary background information and relevant related work. The problem of ranking neural network architectures without training them is an important one, and necessitates more detailed studies.

**Weaknesses:**

I think that the paper has some major points that need to be addressed. Some of these points are listed below:

- The improvements are really marginal compared to methods such as AZ-NAS [1] or Zen-NAS [2], considering there is a much larger relative compute time incurred when using the proposed method in this paper. For instance, in the NB201 search space (Table 3), SABoC-NAS is ~46x more expensive than Zen-NAS. In Table 5, the search costs are actually really high (137h) and the performance is worse than Zen-NAS.

- The benchmarks chosen for evaluation are not enough anymore in my opinion. These search space and datasets have been used for years in NAS research and most of the new proposed methods probably overfit on them. In the future I would suggest designing a method/algorithm on these benchmarks and using other search spaces, tasks, and datasets for an extensive and more comprehensive empirical evaluation.

- The paper does not really contribute much in terms of novelty compared to previously proposed NTK-based zero-cost proxies such as TE-NAS [4] or TEG-NAS [5]. The Borda ranking method is simple and contributes marginally in terms of novelty.

Moreover, I would like to express a general criticism of NTK-based zero-cost proxy methods. It is well-known that using certain parametrizations (e.g. the NTK parameterization), neural networks in the infinite width limit converge in probability to a sample from a Gaussian Process with the NTK prior kernel. However, as He et al. [2020] denote, there is no Bayesian posterior interpretation of trained neural networks. In NAS, we are concerned about the performance of trained neural networks to convergence, and the current NTK-based zero-cost proxies only study the characteristics of a neural network at initialization, without a correct interpretation of the trained NN as a Baysian posterior sample. I would be interested in the authors opinion regarding this point.


-- References --

[1] Lee et al. AZ-NAS: Assembling Zero-Cost Proxies for Network Architecture Search.

[2] Lin et al. Zen-NAS: A Zero-Shot NAS for High-Performance Image Recognition. In ICCV 2021

[3] He et al. Bayesian Deep Ensembles via the Neural Tangent Kernel. In NeurIPS 2020

[4] Chen et al. Neural Architecture Search on ImageNet in Four GPU Hours: A Theoretically Inspired Perspective. In ICLR 2021

[5] Chen et al. Understanding and Accelerating Neural Architecture Search with Training-Free and Theory-Grounded Metrics. In TPAMI 2022

**Questions:**

I would be interested in the following questions to be addressed by the authors:

1. Are such proxies applicable to transformer search spaces, especially in benchmarks such as HW-GPT-Bench [1], which could be seen as  benchmark for structural pruning as well?

2. Can the authors justify the practical relevance of their method considering the substantial search costs (e.g. in Table 5) associated with computing the proposed proxy inside a search routine?


-- References --

[1] Sukthanker et al. "HW-GPT-Bench: Hardware-Aware Architecture Benchmark for Language Models". In NeurIPS 2024, DBT

---

### Official Review · Reviewer_VnU7 · 2024-11-03

**Soundness:** 2
**Presentation:** 2
**Contribution:** 2
**Rating:** 3
**Confidence:** 4

**Summary:**

This paper proposes a new zero-cost NAS method called NTK-score, which leverages the eigenvalues of the NTK or kernel regression to evaluate neural networks based on three key characteristics: trainability, expressivity, and generalization. The paper also introduces SABoC-NAS, a NAS framework that incorporates NTK-score for architecture search. Experimental results demonstrate the effectiveness of both NTK-score and SABoC-NAS.

**Strengths:**

1. The paper introduces a novel zero-shot NAS method by integrating NTK-based metrics to evaluate architectures across trainability, expressivity, and generalization, differing from existing proxies that focus on a single characteristic.
2. The application of NTK provides a solid theoretical foundation for understanding how characteristics of neural networks influence performance.
3. The paper presents experimental results on well-known NAS benchmarks.

**Weaknesses:**

1. The emphasis on trainability may be less significant, as NAS typically prioritizes the final performance of architectures.
2. Results on other popular NAS benchmarks remain unexplored.
3. The tau value presented in Figure 2 is incorrect.
4. The experimental settings for DARTS in this paper are atypical for zero-cost NAS methods, and the results lack competitiveness. Besides, the test accuracy of SABoC-NAS on ImageNet is not reported.

**Questions:**

1. How does SABoC-NAS generalize to other search spaces, such as those in references [1] and [2]?
2. Why is the 100-epoch training accuracy chosen as the auxiliary measure of trainability? A more reasonable approach might be to consider the number of epochs required to achieve the same target accuracy.
3. Which metric is most important when evaluating an architecture? It appears that SABoC-NAS assigns equal weight to all three metrics according to Equation 12.

[1] NAS-Bench-101: Towards Reproducible Neural Architecture Search.
[2] TransNAS-Bench-101: Improving Transferability and Generalizability of Cross-Task Neural Architecture Search.

---

### Official Review · Reviewer_9mDb · 2024-11-04

**Soundness:** 3
**Presentation:** 3
**Contribution:** 2
**Rating:** 5
**Confidence:** 4

**Summary:**

The Neural Tangent Kernel (NTK) is a theoretical framework for understanding and analyzing various characteristics of neural networks. Consequently, it is one of the commonly-used frameworks to anticipate the performance of randomly-initialized neural networks in Neural Architecture Search. This paper introduces a new NTK-score, which is designed to simultaneously evaluate the network's trainability, expressivity, and generalization. The trainability is measured through the eigenvalues of the NTK matrix, the expressivity through NTK-based kernel regression, and the generalization through the square loss between the estimated and the true label. Based. on the NTK-score, the authors employ the Borda Count approach to balance to trade-off among three characteristics. The proposed score is verified on NAS-Bench-201 from the perspective of Neural Architecture Search and network pruning. It can also be used with reinforce and evolutionary algorithms.

**Strengths:**

- The paper is well-written and easy to follow.

- Although the paper does reuse some of the ideas from previous literature (i.e., metrics for trainability and generalization), it does introduce a new kernel regression-based metric to measure expressivity. Also, this is the first work to utilize the Borda count approach to take multiple aspects of neural networks' characteristics into account.

- The effectiveness of the proposed method is validated on two different applications: Neural Architecture Search and Network Pruning.

**Weaknesses:**

- As mentioned earlier, a significant portion of the proposed method is borrowed from the existing literature. In that sense, I believe utilizing the Borda count algorithm to conduct architecture search is really the only technical contribution of this paper.

- Unlike the proposed method, many of the other zero-cost proxies utilizes a single metric to search for the optimal architecture. Then, can these previously-proposed proxies simply be combined into a single metric and used together to conduct architecture search through Borda count? I would like to see this additional result in the rebuttal.

- Is this method robust to changes in the search space, weight initialization, data domains (e.g., image/text/speech)? NAS-Bench-201 is one of the more restrictive search spaces. I would like to see whether this method can stably perform on other more comprehensive search spaces.

- To truly show that this metric searches for a network with better generalization ability, shouldn't the searched architecture also be verified in various out-of-distribution scenarios? (e.g., testing the network searched on CIFAR-10 on CIFAR-10-c or -p?)

**Questions:**

Please refer to the weaknesses section.

---

### Note · Authors · 2025-01-16

I have read and agree with the venue's withdrawal policy on behalf of myself and my co-authors.